

# re-Searcher: GUI-based bioinformatics tool for simplified genomics data mining of VCF files

Daniyar Karabayev[1,*], Askhat Molkenov[1], Kaiyrgali Yerulanuly[1,2], Ilyas Kabimoldayev[1], Asset Daniyarov[1], Aigul Sharip[1], Ainur Ashenova[1], Zhaxybay Zhumadilov[1,3] and Ulykbek Kairov[1,*]

[1] Laboratory of Bioinformatics and Systems Biology, Center for Life Sciences, National Laboratory Astana, Nazarbayev University, Nur-Sultan, Kazakhstan
[2] L.N. Gumilyov Eurasian National University, Nur-Sultan, Kazakhstan
[3] School of Medicine, Nazarbayev University, Nur-Sultan, Kazakhstan
[*] These authors contributed equally to this work.

## ABSTRACT

**Background**. High-throughput sequencing platforms generate a massive amount of high-dimensional genomic datasets that are available for analysis. Modern and user-friendly bioinformatics tools for analysis and interpretation of genomics data becomes essential during the analysis of sequencing data. Different standard data types and file formats have been developed to store and analyze sequence and genomics data. Variant Call Format (VCF) is the most widespread genomics file type and standard format containing genomic information and variants of sequenced samples.

**Results**. Existing tools for processing VCF files don't usually have an intuitive graphical interface, but instead have just a command-line interface that may be challenging to use for the broader biomedical community interested in genomics data analysis. re-Searcher solves this problem by pre-processing VCF files by chunks to not load RAM of computer. The tool can be used as standalone user-friendly multiplatform GUI application as well as web application (https://nla-lbsb.nu.edu.kz). The software including source code as well as tested VCF files and additional information are publicly available on the GitHub repository (https://github.com/LabBandSB/re-Searcher).

# INTRODUCTION

Recent achievements in high-throughput sequencing technologies (*Goodwin, McPherson & McCombie, 2016*; *van Dijk et al., 2018*) have led to the generation of massive amounts of genomic data (*Gao et al., 2019*; *The ICGC/TCGA Pan-Cancer Analysis of Whole Genomes Consortium, 2020*) available for the research community. Many omics databases have been developed and have collected freely accessible datasets (*Molkenov et al., 2019*; *Rigden & Fernández, 2020*) for analysis by the bioinformatics community. Modern bioinformatics tools and methods are in high demand for analyzing and interpreting the big omics data generated by the different types of multi-omics platforms available. Different

Corresponding author
Ulykbek Kairov,
ulykbek.kairov@nu.edu.kz

standard data types and file formats have been developed to store and analyze sequence and genomics data. Variant Call Format (VCF) (*Danecek et al., 2011*) is a tab-delimited text file format that is often used in bioinformatics to store genomic variants. A VCF file consists of the header, including meta-information lines and field definition lines (column names), and the body (data section). An arbitrary number of meta-information lines start with '##' and provide a description of the VCF file. The body of the file consists of eight mandatory columns: chromosome (CHROM), starting position of a variant (POS), variant identifiers (ID), the reference allele (REF), a list of alternate alleles (ALT), a PHRED-scaled quality score (QUAL), filter information regarding variant validity (FILTER), and annotation information (INFO). Additional columns describing samples can also be added. Each row of the file describes specific genomic variants (SNVs, INDELs, CNVs, and other structural variants) at the given chromosome and genomic position.

VCF files often store information about numerous samples and can therefore reach huge sizes–gigabytes, or sometimes terabytes. This creates an issue for the readability of VCF files and further analysis for non-programmers, as manual data extraction and analysis using Microsoft Excel or other table processing software may not be possible due to the RAM capacity limitation of standard computers. We introduce re-Searcher, bioinformatics tool specifically developed for simplified mining and analysis of big-size VCF files. We developed a multi-platform user-friendly graphical user interface (GUI) tool for offline access, while re-Searcher web application can be used online via web browser. re-Searcher has been developed for the broader biomedical community and solves the problem of working and analyzing genomics data stored in VCF format.

## MATERIALS & METHODS

### Implementation

The re-Searcher application was written in Python 3 (*Van Rossum & Drake, 2010*) with the implementation of the Tkinter (*Python Software Foundation, 2020*) package to build the GUI, and the Pandas (*McKinney, 2010*) library to extract columns. For convenience of users who would like to build re-Searcher into their pipelines command line interface (CLI) is available as python script (see Availability). CLI mirrors functionality of GUI regarding files processing. We developed web version of re-Searcher for users to manipulate VCF files without downloading CLI or GUI versions of re-Searcher. The web application was developed using Django web framework (*Django Software Foundation, 2013*), to run python script via WSGI on Apache web server (*Fielding & Kaiser, 1997*). The web version runs re-Searcher scripts on server and takes input files from website and returns processed files to user to download.

re-Searcher solves the problem of analyzing large VCF files by not loading the whole file directly into RAM, but instead pre-processing it in chunks and utilizing a simple and intuitive interface (Fig. 1). The main advantage of re-Searcher in comparison with other tools is the presence of a simple and user-friendly interface, GUI and web interface, instead of a CLI, as well as a lack of confused installation procedures typical for existing tools. The generalized workflow of re-Searcher consists of several steps: selecting an input file, setting
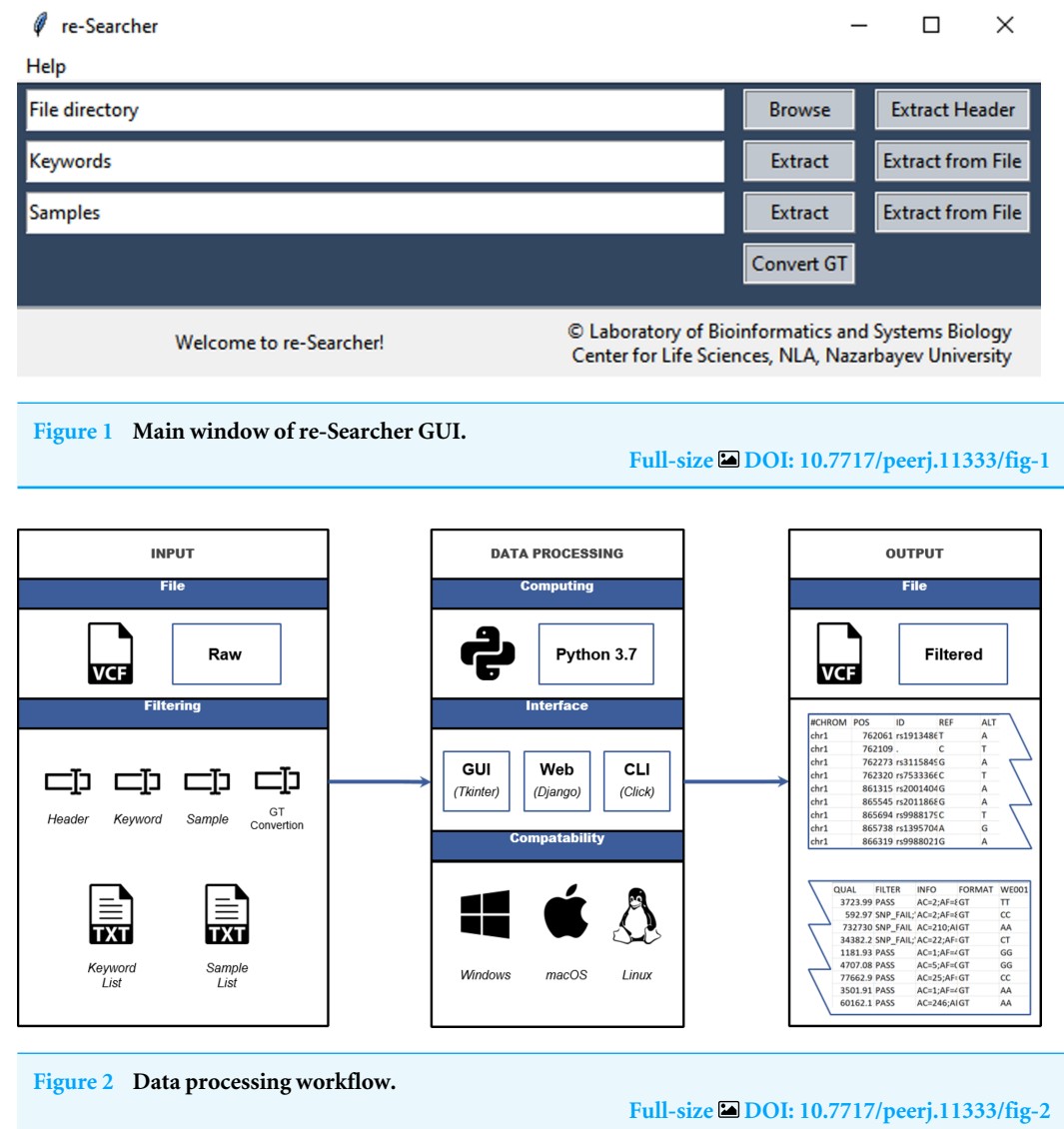

**Figure 1** Main window of re-Searcher GUI.

**Figure 2** Data processing workflow.

up necessary filtering parameters, data processing, and exporting a filtered output VCF file (Fig. 2). re-Searcher browses and opens VCF files with extensions ".txt" or ".vcf", before performing the following filtering and extraction options:

## Header extraction

VCF files can be large and, if a user needs to know only certain information in a file header (e.g., a particular meta-line or sample ID), the software can extract only the header from the original VCF file and save it into a new file.

## Keyword search

If genomic variants need to be filtered according to the presence of a keyword, the software can find these rows and extract them into a new VCF file. Users may input multiple keywords by accessing the entry field or by uploading a .txt file with keywords.

**Figure 3** **Genotype conversion example.** (A) The numeric genotype of biallelic and multiallelic variants before conversion, and (B) letter genotype of the same variants after conversion.

## Sample extraction

If the user needs only particular sample IDs in a VCF file, the software can extract the necessary sample columns into a new file. Similar to a keyword search, users may input multiple sample IDs by accessing the entry field or by uploading a .txt file with the IDs.

## Genotype format conversion

re-Searcher can convert numeric genotype (GT) format into letter format. The conversion option is one of the most used operations when working with VCF files, for example, in further comparative analysis of genetic variants or SNPs. The original GT format in VCF files is numeric (*0/0, 0/1, 1/1* for biallelic sites or *1/2, 2/3*, etc. for multiallelic sites), where *0* is a reference (REF) allele, *1* is a first alternative (ALT) allele, *2* is a second ALT allele and so on (Danecek et al., 2011; (*Campbell et al., 2016*). After GT format conversion REF number is replaced with REF letter and ALT number with corresponding ALT letter (Fig. 3). For instance, if GT of first sample is *0/1* and GT of second sample is *1/2* in numeric format, while REF and two ALT are *GC, T* and *CAA* respectively, then after conversion first sample's letter GT becomes *GC, T* and second sample's letter GT becomes *T, CAA*.

The final output file is a filtered and processed VCF file and is generated with a complement log file containing the file processing information, name of specified file and work directory with outputs.
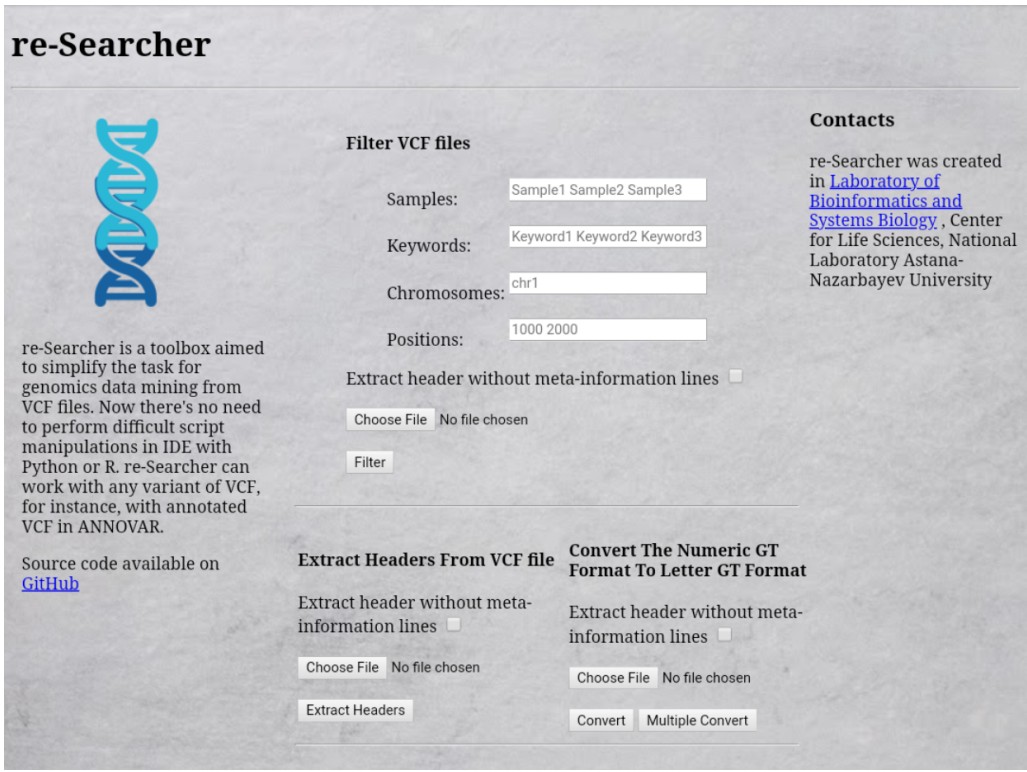

**Figure 4** Web interface of re-Searcher available via browser at https://nla-lbsb.nu.edu.kz.

## RESULTS AND DISCUSSION

We have compared the main features of re-Searcher with other existing and open source tools VIVA (*Tollefson et al., 2019*), VCFtools (*Danecek et al., 2011*), GEMINI (*Paila et al., 2013*), BrowseVCF (*Salatino & Ramraj, 2016*), VCF.Filter (*Müller et al., 2017*), VCF-Miner (*Hart et al., 2016*) and prepared a detailed table (Table 1). The compared tools have been developed and implemented in different programming languages (Julia, C++, Perl, Java and Python) and dedicated libraries. VCFtools is one of the most cited and advanced tools for processing VCF files, but it requires additional computational skills for effective usage. VIVA is the only tool that provides the possibility for advanced visualization and plotting figures. In addition to re-Searcher, other tools with a GUI available for users are BrowseVCF, VCF.Filter, and VCF-Miner, while web interface is available in only in re-Searcher (Fig. 4) and GEMINI. From these, only re-Searcher, BrowseVCF and VCF.Filter tools support multiple operational systems (Windows, MacOS, Linux). Searching the whole VCF file by keyword and the corresponding extraction of data based on keywords are features available on re-Searcher and BrowseVCF, whereas genotype conversion is a unique feature of re-Searcher.

re-Searcher is a multi-platform tool and can be run on MacOS, Windows and Linux operating systems. Performance of re-Searcher has been evaluated on these PC platforms (Windows 10 Pro OS and Linux Mint 17 OS-based PC: CPU Intel Core i5-8250U 1.80

Karabayev et al. (2021), *PeerJ*, DOI 10.7717/peerj.11333

**Table 1 Comparison of re-Searchers features with similar tools.**

| Categories | Features | re-Searcher | VIVA | VCFtools | GEMINI | BrowseVCF | VCF.Filter | VCF-Miner |
|---|---|---|---|---|---|---|---|---|
| Technical Aspects | Compatibility with operation system | Windows, MacOS, Linux | Windows, MacOS, Linux | Windows, MacOS, Linux | Windows, MacOS, Linux | Windows, MacOS, Linux | Windows, MacOS, Linux | Windows |
| | Language | Python | Julia | C++, Perl | Python | Python, JavaScript, CSS, HTML5 | Java | Java |
| | Interface | GUI, Web Browser, CLI | CLI, Jupyter Notebook | CLI | Web Browser, CLI | GUI, CLI | GUI | GUI |
| | Works offline | V | V | V | X | X | V | V |
| | Portable launcher | V | X | X | X | V | X | X |
| Functionality | Search by keyword | V | X | X | X | V | X | X |
| | Sample selection | V | V | V | V | V | V | V |
| | Genotype format conversion | V | X | X | X | X | X | X |
| | Visualization | X | V | X | X | X | X | X |
| | Export filtered VCF file | V | X | V | X | X | V | V |

**Table 2  re-Searcher multi-platform run time comparison.**

| OS | VCF file size (Gb) | Execution Time (sec) | | | |
|---|---|---|---|---|---|
| | | Extract header | Keyword extraction | Sample ID extraction | GT conversion |
| Linux | 0.081 | 2.227 | 9.995 | 15.243 | 38.375 |
| | 0.814 | 2.497 | 38.106 | 21.334 | 117.186 |
| | 1.320 | 2.462 | 67.260 | 61.159 | 206.868 |
| | 1.980 | 2.115 | 75.331 | 104.167 | 330.527 |
| | 7.950 | 6.145 | 366.137 | 200.347 | 1117.168 |
| Windows | 0.081 | 14.865 | 22.482 | 4.785 | 49.642 |
| | 0.814 | 18.898 | 48.820 | 21.398 | 139.641 |
| | 1.320 | 21.054 | 44.669 | 59.329 | 238.192 |
| | 1.980 | 10.919 | 53.958 | 90.446 | 339.275 |
| | 7.950 | 16.308 | 502.996 | 309.177 | 1320.197 |
| MacOS | 0.081 | 9.627 | 9.297 | 10.423 | 16.298 |
| | 0.814 | 5.262 | 20.916 | 19.705 | 116.82 |
| | 1.320 | 6.286 | 35.433 | 53.247 | 186.181 |
| | 1.980 | 3.231 | 37.923 | 119.136 | 286.544 |
| | 7.950 | 5.457 | 148.612 | 254.128 | 1130.225 |

**Notes.**

OS, operational system; GT, genotype; sec, seconds; Gb, gigabyte.

GHz, RAM 4Gb and MacOS Catalina based PC: CPU Intel Core i7, 3.2 GHz, RAM 8 Gb) with different VCF file sizes. Different size VCF files (0.081 Gb, 0.814 Gb, 1.320 Gb, 1.980 Gb and 7.950 Gb) were used as input datasets for evaluating re-Searcher performance. The results of the performance benchmarking are shown in Table 2.

We have used big VCF files from the 1000 Genomes Project (*The 1000 Genomes Project Consortium, 2015*) and then generated the different sized testing VCF files from this dataset. The Linux-based systems had the fastest execution time for different operations in comparison with Windows and MacOS systems.

## Availability

re-Searcher executable software including source code, tested VCF files and additional information are publicly available on the GitHub repository https://github.com/LabBandSB/re-Searcher. re-Searcher is free bioinformatics tool and open to all users without login and registration requirements and do not require an installation of additional tools. CLI version of re-Searcher is also available on the GitHub repository for incorporation into other pipelines. In addition, web version of re-Searcher is available at https://nla-lbsb.nu.edu.kz.

## CONCLUSIONS

Exploring and analyzing VCF files generated after the bioinformatics processing of sequencing data is one of the important steps performed by researchers during analysis and meta-analysis of genotype/phenotype associations. We have developed and introduced an easy-to-use bioinformatics tool, re-Searcher, with several unique features for mining

big VCF files and realized with a simple graphical user interface and web interface that makes it easily available for clinicians and researchers without any computational skills. Several improvements such as visualization options (clustering and plotting functions) with Principal Component Analysis and heatmap methodologies are under future development of re-Searcher.

## ACKNOWLEDGEMENTS

This work is dedicated to the blessed memory of Dr. Vasily Ogryzko.

### Funding

This work has been supported by grant projects (AP08855353, AP05135430 and AP05136106) and program-targeted funding (PTF No. BR05236508) from the Committee of Science, Ministry of Education and Science of the Republic of Kazakhstan. The funders had no role in study design, data collection and analysis, decision to publish, or preparation of the manuscript.

### Grant Disclosures

The following grant information was disclosed by the authors:
Committee of Science, Ministry of Education and Science of the Republic of Kazakhstan: AP08855353, AP05135430, AP05136106, PTF No. BR05236508.

### Competing Interests

The authors declare there are no competing interests.

### Author Contributions

- Daniyar Karabayev conceived and designed the experiments, performed the experiments, prepared figures and/or tables, authored or reviewed drafts of the paper, and approved the final draft.
- Askhat Molkenov conceived and designed the experiments, analyzed the data, prepared figures and/or tables, and approved the final draft.
- Kaiyrgali Yerulanuly, Ilyas Kabimoldayev, Asset Daniyarov, Aigul Sharip and Ainur Ashenova performed the experiments, prepared figures and/or tables, and approved the final draft.
- Zhaxybay Zhumadilov conceived and designed the experiments, authored or reviewed drafts of the paper, supervised the study and were involved in funding acquisition, and approved the final draft.
- Ulykbek Kairov conceived and designed the experiments, performed the experiments, analyzed the data, prepared figures and/or tables, authored or reviewed drafts of the paper, supervised the study and were involved in funding acquisition, and approved the final draft.

## Data Availability

The software including source code and tested VCF files and additional information are available at GitHub: https://github.com/LabBandSB/re-Searcher.

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
