# Peer review of "re-Searcher: GUI-based bioinformatics tool for simplified genomics data mining of VCF files"

_PeerJ, doi:10.7717/peerj.11333_

## Round 0.1 · original submission · Major Revisions

The reviewers have raised some interest in this work and have suggested that the tool can be useful for those without a technical background in bioinformatics. However, they have challenged the tool's utility for technical users with limited advantage against available tools, which warrants a discussion on comparison with available pipelines. The reviewers also have raised some concerns about how the pipeline has been made available. Following the reviewers' comments, I recommend the inclusion of the web-based tool in revision and providing a guide and a few additional examples for the tool, so its different usages can be demonstrated to a broader audience, especially to those who are new to the field.

·

Basic reporting

This manuscript was very well structured and written.

Experimental design

The primary aim of the project was clearly defined and meaningful. The project provides a new GUI tool for working with VCF files. The methods description was clearly and the tool designed is easily accessible at github.

Validity of the findings

Authors designed and wrote this GUI-base bioinformatics tool for mining VCF files. The tool designed is easily accessible at github and the tests work.

Additional comments

In this manuscript the authors described a GUI-base bioinformatics tool for simplified genomics data mining of VCF files. This tool would be a great help to researchers lacking command line computation skills. Under Materials & Methods, the authors described how to convert genotype format (line 86). It seems the authors did not discuss multiallelic variants (https://gatk.broadinstitute.org/hc/en-us/articles/360035890771-Biallelic-vs-Multiallelic-sites#:~:text=A%20multiallelic%20site%20is%20a,two%20or%20more%20variant%20alleles.), which are common in cancer sample. Additionally on the tool's github website, the authors seemed to label multiple nucleotide variants as multiallelic variants (under Features section, item 5, bottom graph), which could confuse and mislead potential users.

Reviewer 2 ·

Basic reporting

This is a paper about a tool to visualize vcf files. Authors explained the tool in a clear manner in the manuscript. Existing tools were mentioned and compared. Figures and table are clear.
Here are some suggestions for the authors:

1. The reviewer looked at the github for the tool and first couldn't access as the link provided in line 122 is incorrect. The link provided in the abstract is correct.
2. The reviewer did not find an .exe file in the github although it was mentioned in the manuscript that the tool works with .exe file. The code is available, but for the users who do not have any background in programming or bioinformatics this could be very problematic. The authors need to make sure their github page is user friendly for all users with diverse backgrounds as they mentioned that this tool would be useful for a "broader biomedical community".
3. Please include clear instructions for installation. No instructions for installation was found in the github page.

Experimental design

The authors mentioned that they are working on a web-based tool. The reviewer suggests to include the web-based tool in this version if possible. It would be very useful for the users with no programming background.

Validity of the findings

No comment

Reviewer 3 ·

Basic reporting

Authors used professional english and also cited the literature references wherever needed.

Experimental design

This is not an original research article. There are already various other software available for this purpose.

Validity of the findings

I don't think this is a novel research idea and will not make any further impact and neither will be more useful for researchers than previously published software.

Additional comments

In this manuscript, authors developed a software called "re-Searcher" to process and visualize VCF files, which is not giving any additional advantage over previously published software. Only unique functionality of "re-Searcher" is to convert numerical genotypes to alphabetical genotypes, which is also not sufficient to claim a manuscript.

---

## Round 0.2 · accepted · Accept

Congratulations on the acceptance of your paper! As a reviewer noted, this tool will be potentially helpful for researchers without expertise in bioinformatics.

·

Basic reporting

This manuscript was very well structured and written.

Experimental design

The primary aim of the project was clearly defined and meaningful. The project provides a new GUI tool for working with VCF files. The methods description was clearly and the tool designed is easily accessible at github.

Validity of the findings

Authors designed and wrote this GUI-base bioinformatics tool for mining VCF files. The tool designed is easily accessible at github and the test exampless work.

Additional comments

In this manuscript the authors described a GUI-base bioinformatics tool for simplified genomics data mining of VCF files. This tool would be a great help to researchers lacking command line computation skills. The revision addressed my concerns from the previous version.

Reviewer 2 ·

Basic reporting

The authors responded to all comments well.

Experimental design

The webpage with the user interface looks good. The github page has been improved.

Validity of the findings

No comment